# Postoperative effects of laparoscopic Hartmann reversal: A multicenter propensity score matched study

**Kil-yong Lee[1], Jaeim Lee** **[1]\*, Seong Taek Oh[1], Chul Seung Lee[2], Nam Suk Kim[3], Ju Myung Song[4], Ri-Na Yoo[5], Byung Jo Choi[6]**

**1** Department of Surgery, Uijeongbu St. Mary's Hospital, College of Medicine, The Catholic University of Korea, Uijeongbu-si, Republic of Korea, **2** Department of Surgery, Seoul St. Mary's Hospital, College of Medicine, The Catholic University of Korea, Seoul, Republic of Korea, **3** Department of Surgery, Yeouido St. Mary's Hospital, College of Medicine, The Catholic University of Korea, Seoul, Republic of Korea, **4** Department of Surgery, Incheon St. Mary's Hospital, College of Medicine, The Catholic University of Korea, Incheon, Republic of Korea, **5** Department of Surgery, St. Vincent's Hospital, College of Medicine, The Catholic University of Korea, Suwon, Republic of Korea, **6** Department of Surgery, Daejeon St. Mary's Hospital, College of Medicine, The Catholic University of Korea, Daejeon, Republic of Korea

\* lji96@catholic.ac.kr

## Abstract

### Background

Although the advantages of laparoscopic Hartmann reversal (LHR) compared to open Hartmann reversal (OHR) have been reported in the literature, the number of multicenter studies with good matching investigating this topic is rare. In the present study, we aimed to confirm the advantages of LHR in terms of short-term outcomes through propensity score matching of LHR and OHR groups, using data collected from multiple institutions.

### Methods

Patients who underwent Hartmann reversal at six institutions under the Catholic Medical Center of the Catholic University of Korea between January 1, 2005, and December 31, 2021, were included. The patients were divided into the LHR and OHR groups based on the technique used. The two groups were matched using propensity score matching (1:1 ratio, logistic regression with the nearest-neighbor method). The primary outcome was postoperative ileus (POI) frequency, and secondary outcomes were time to solid diet (days) and length of stay (days).

### Results

Among 337 patients, propensity score matching was performed on 322, after excluding 15 who had undergone open conversion. Of these, 63 patients were assigned to each group through propensity score matching. There was no difference in the frequency of adhesiolysis (77.8% vs. 82.5%, p = 0.503) or the operation time. (210 (IQR 159–290) vs. 233 (IQR 160–280), p = 0.718) between the two groups. As the primary outcome, the LHR group showed significantly lower POI frequency than the OHR group. (4.8% vs. 22.2%, p =

irbujb@catholic.ac.kr) for researchers who meet the criteria for access to confidential data.

**Funding:** The author(s) received no specific funding for this work.

**Competing interests:** The authors have declared that no competing interests exist.

0.0041) Regarding the secondary outcomes, the LHR group showed a shorter period to solid diet than the OHR group. The length of hospital stay was also significantly shorter in the LHR group (4 vs. 6, p < 0.0001; 9 vs. 12, p<0.0001).

## Conclusion

LHR is an effective method to ensure faster recovery of patients after surgery compared to OHR.

## Introduction

Laparoscopic colorectal surgery for colorectal disease has several advantages, including a reduction in both postoperative pain and hospital stay [1–3]. Furthermore, even highly difficult laparoscopic surgeries such as those of adhesive small bowel obstruction can be safely performed [4]. Hartmann's operation is performed when primary anastomosis of the colon is difficult due to severe peritonitis caused by left colonic perforation or unstable vital signs [5], and Hartmann reversal is often performed in some patients who have undergone Hartmann's procedure. However, there are cases in which surgeons hesitate to use laparoscopic surgery because of the difficulty of detachment due to peritoneal adhesions caused by a history of previous surgery [6–10]. However, it has been reported that laparoscopic surgery can be performed safely even with single incision during Hartmann reversal [5, 11, 12]. Nevertheless, there is debate regarding the merits of laparoscopic Hartmann's reversal over open procedures due to the lack of randomized controlled studies.

To address this knowledge gap, in the present study, we aimed to confirm the advantages of laparoscopic Hartmann reversal (LHR) by confirming the difference in postoperative complications using propensity score matching in two groups of patients (laparoscopic vs open approaches) among patients undergoing Hartmann reversal at several centers.

## Methods

This study is a multicenter retrospective cohort study. This study was approved by the institutional review board (IRB) of the Catholic University of Korea and was performed in accordance with the IRB's guidelines and regulations. The requirement for informed consent was waived by the IRB.

### Patients

Patients who underwent Hartman's reversal at six institutions under the Catholic Medical Center of the Catholic University of Korea between January 1, 2005 and December 31, 2021, were included. The exclusion criteria were conversion and missing data for the covariates used in propensity score matching. All the data were retrospectively reviewed.

According to the research results of Ng et al [13] and Yang et al [14], when the sample size was calculated with 80% power and alpha 0.05, the minimum sample size required for each group was 60 patients.

### Definition

Postoperative ileus (POI) was defined as any situation that requires a return to "nil per os" or the insertion of a nasogastric tube (NG) [15]. Stump length was measured using

sigmoidoscopy prior to surgery. Anastomotic stricture was defined as the inability to pass a 13.2-mm colonoscope through the colon, or the feeling of resistance during its passage [16].

### Outcomes

The primary outcome was POI frequency, and the secondary outcomes were time to solid diet (days), length of hospital stay (days), and postoperative complications.

### Statistical analysis

Patients were divided into two groups: the LHR and the open Hartmann reversal (OHR) groups for comparison of the clinical characteristics. Continuous variables were assessed using the independent t-test or Wilcoxon rank sum test, and discontinuous variables were analyzed using the Chi-square test and Fisher exact test.

Propensity score matching (1:1 ratio using logistic regression with the nearest-neighbor method) was applied to correct for factors that differed between the two groups. The covariates were age, sex, body mass index, smoking, diabetes, hypertension, heart disease, pulmonary disease, liver disease, cerebrovascular disease, stump length, cause of perforation (cancer or benign), combined resection, anastomosis method, and stapler size.

Statistical analyses were performed using SAS ver 9.4 (SAS Institute Inc., Cary, NC, USA). Statistical significance was set at $p < 0.05$.

## Results

Of the 337 patients who underwent Hartmann's reversal, 322 were included in this study, after excluding 15 who underwent conversion. Among the 322 included patients, 89 underwent LHR and 233 underwent OHR (Fig 1). The baseline characteristics are shown in Table 1. After propensity score matching, 63 participants were assigned to each group; their baseline characteristics are shown in Table 2. The post-matching patient number of this study was shown to be 82% in power analysis, which was an appropriate sample size.

### Operation related factors

There were no differences in terms of stump length ($p = 0.925$), adhesiolysis ($p = 0.503$), anastomotic method ($p = 0.803$), diversion ($p = 0.492$), or surgery time ($p = 0.718$) between the two groups (Table 3).

### Primary outcome

The POI frequencies with prevalence in the LHR and OHR groups were 3 (4.76%) and 14 (22.2%), respectively, showing significant differences ($p = 0.004$) (Table 4).

In the LHR group, two POI patients improved symptoms 2 and 3 days after L-tube insertion with parenteral nutritional support, respectively. However one patient underwent laparoscopic adhesiolysis with transverse colectomy to resolve adhesions.

Of the 13 POI patients in the OHR group, 3 patients spontaneously improved their symptoms without L-tube insertion. However, 1 patient underwent open adhesiolysis with T-colostomy to resolve the adhesion.

### Secondary outcomes

The median lengths of stay in the LHR and OHR groups were 9 and 12 days, respectively, showing a significantly shorter length of stay in the LHR group ($p < 0.001$). In terms of median time to solid diet, the LHR group showed a significantly shorter period than the OHR group (4

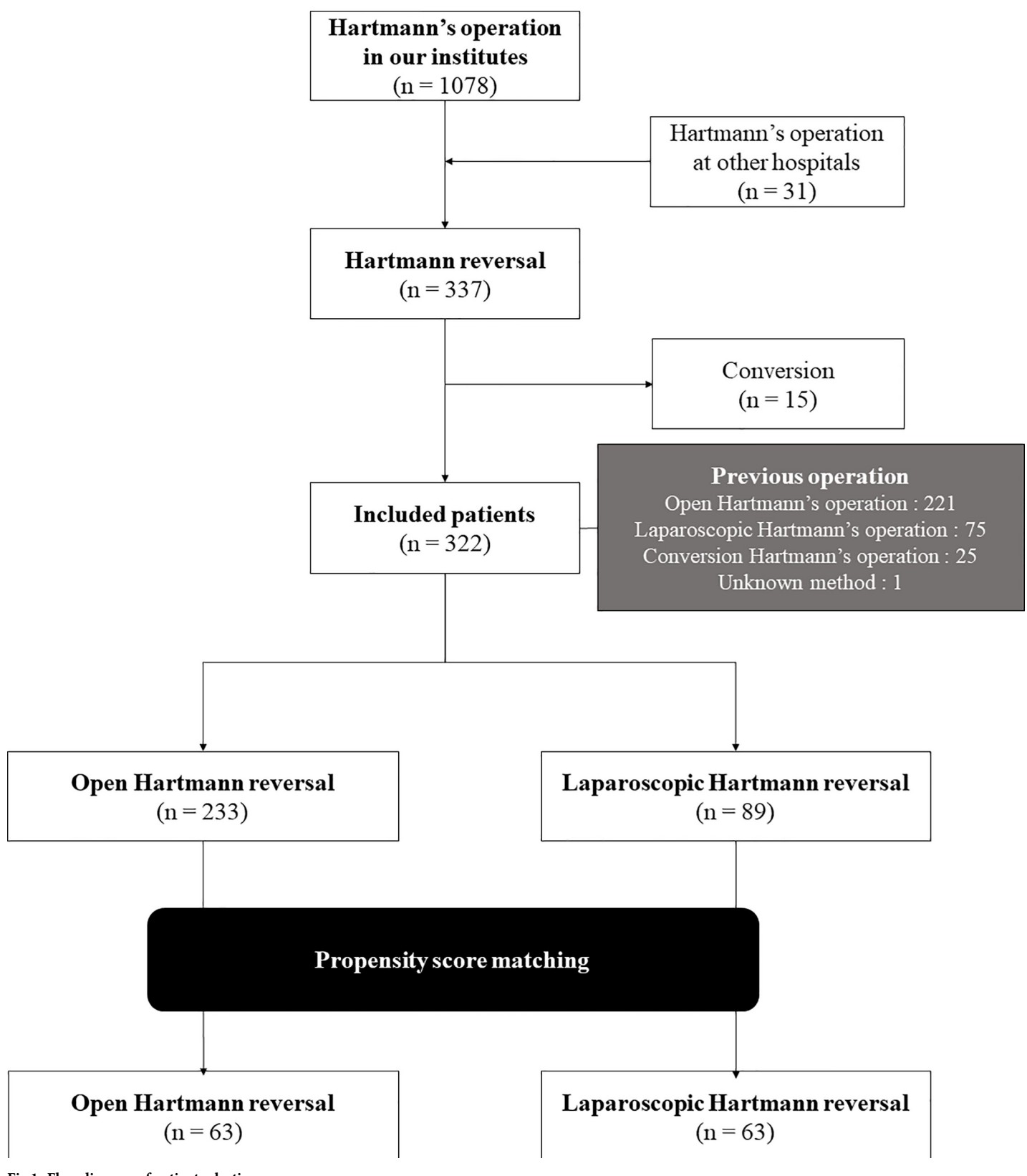

**Fig 1. Flow diagram of patient selection.**

**Table 1. Baseline characteristics before propensity score matching.**

| Variables | | Baseline characteristics before Propensity Score Matching | | |
|---|---|---|---|---|
| | | LHR (n = 89) | OHR (n = 233) | p-value |
| Age (years) | | 69 (IQR 59–75) | 66.5 (IQR 57–74) | 0.2300(W) |
| Sex | Male | 45 (50.56%) | 112 (48.07%) | 0.6890(C) |
| | Female | 44 (49.44%) | 121 (51.93%) | |
| Height (cm) | | 160 (IQR 151.2–165.2) | 157.6 (IQR 151.7–165.0) | 0.7861(W) |
| Weight (kg) | | 58 (IQR 50.6–65.3) | 58.7 (IQR 52.0–65.6) | 0.8561(W) |
| Body mass index (kg/m$^2$) | | 23.2 (IQR 20.9–25.8) | 23.4 (IQR 21.2–25.5) | 0.8027(W) |
| Smoking | No | 76 (85.39%) | 192 (82.40%) | 0.6819(C) |
| | Past | 5 (5.62%) | 12 (5.15%) | |
| | Present | 8 (8.99%) | 29 (12.45%) | |
| ASA classification | 1 | 9 (10.23%) | 33 (14.16%) | 0.6119(C) |
| | 2 | 68 (77.27%) | 169 (72.53%) | |
| | 3 | 11 (12.50%) | 31 (13.30%) | |
| Diabetes | Yes | 16 (17.98%) | 45 (19.31%) | 0.7844(C) |
| | No | 73 (82.02%) | 188 (80.69%) | |
| Hypertension | Yes | 54 (60.67%) | 127 (54.51%) | 0.3185(C) |
| | No | 35 (39.33%) | 106 (45.49%) | |
| Heart disease | Yes | 16 (17.98%) | 31 (13.30%) | 0.2882(C) |
| | No | 73 (82.02%) | 202 (86.70%) | |
| Pulmonary disease | Yes | 12 (13.48%) | 20 (8.58%) | 0.1888(C) |
| | No | 77 (86.52%) | 213 (91.42%) | |
| Liver disease | Yes | 3 (3.37%) | 6 (2.58%) | 0.7113(F) |
| | No | 86 (96.63%) | 227 (97.42%) | |
| Cerebrovascular disease | Yes | 10 (11.24%) | 11 (4.72%) | 0.0342*(C) |
| | No | 79 (88.76%) | 222 (95.28%) | |
| Cancer or benign | Cancer | 31 (34.83%) | 80 (34.33%) | 0.9332(C) |
| | benign | 58 (65.17%) | 153 (65.67%) | |
| Hartmann method | Open | 19 (21.35%) | 202 (87.07%) | <0.0001*(C) |
| | Laparoscopy | 66 (74.16%) | 9 (3.88%) | |
| | Conversion | 4 (4.49%) | 21 (9.05%) | |

LHR, Laparoscopic Hartmann Reversal; OHR, Open Hartmann Reversal; ASA, American Society of Anesthesiologists; IQR, interquartile range

p-value: Independent t-test (T) or Wilcoxon rank sum test (W)

p-value: Chi-square test (C) or Fishers exact test (F)

vs 6 days, p<0.001). There was no difference in postoperative complications between the two groups (Table 4).

## Subgroup analysis

Of the 246 patients who underwent open or conversion Hartmann's operation in included 322 patients (Fig 1), 223 had undergone OHR and 23 had undergone LHR; the frequency with prevalence of POI between the two groups was 1 (4.4%) for LHR and 48 (21.5%) for OHR (p = 0.055). In terms of median time to solid diet, the LHR group (5 days) showed a significantly faster diet than the OHR group (6 days) (p<0.001). The median lengths of stay in the LHR and OHR groups were 9 and 12 days, respectively, showing a significantly shorter length of stay in the LHR group (p<0.001). Complications did not differ between the two groups (S4 Table).

**Table 2. Baseline characteristics after propensity score matching.**

| Variables | | Baseline characteristics after Propensity Score Matching | | |
|---|---|---|---|---|
| | | LHR (n = 63) | OHR (n = 63) | p-value |
| Age (years) | | 70 (IQR 62–75) | 67 (IQR 57–75) | 0.5695(W) |
| Sex | Male | 30 (47.62%) | 28 (44.44%) | 0.7207(C) |
| | Female | 33 (52.38%) | 35 (55.56%) | |
| Height (cm) | | 158 (IQR 150.1–165.0) | 156 (IQR 149.7–165.0) | 0.8224(W) |
| Weight (kg) | | 58.73 ± 12.54 | 58.88 ± 10.83 | 0.9442(T) |
| Body mass index (kg/m$^2$) | | 23.3 (IQR 20.9–25.8) | 23.4 (IQR 21.6–24.7) | 0.9455(W) |
| Smoking | No | 57 (90.48%) | 58 (92.06%) | 0.5784(F) |
| | Past | 2 (3.17%) | 0 (0.00%) | |
| | Present | 4 (6.35%) | 5 (7.94%) | |
| ASA classification | 1 | 7 (11.11%) | 7 (11.11%) | 0.9688(C) |
| | 2 | 47 (74.60%) | 46 (73.02%) | |
| | 3 | 9 (14.29%) | 10 (15.87%) | |
| Diabetes | Yes | 10 (15.87%) | 10 (15.87%) | 1.0000(C) |
| | No | 53 (84.13%) | 53 (84.13%) | |
| Hypertension | Yes | 39 (61.90%) | 41 (65.08%) | 0.7113(C) |
| | No | 24 (38.10%) | 22 (34.92%) | |
| Heart disease | Yes | 12 (19.05%) | 9 (14.29%) | 0.4733(C) |
| | No | 51 (80.95%) | 54 (85.71%) | |
| Pulmonary disease | Yes | 6 (9.52%) | 4 (6.35%) | 0.5098(C) |
| | No | 57 (90.48%) | 59 (93.65%) | |
| Liver disease | Yes | 1 (1.59%) | 1 (1.59%) | 1.0000(F) |
| | No | 62 (98.41%) | 62 (98.41%) | |
| Cerebrovascular disease | Yes | 7 (11.11%) | 2 (3.17%) | 0.1638(F) |
| | No | 56 (88.89%) | 61 (96.83%) | |
| Cancer or benign | Cancer | 22 (34.92%) | 18 (28.57%) | 0.4440(C) |
| | benign | 41 (65.08%) | 45 (71.43%) | |
| Hartmann method | Open | 7 (11.11%) | 60 (95.24%) | <0.0001*(F) |
| | Laparoscopy | 53 (84.13%) | 1 (1.59%) | |
| | Conversion | 3 (4.76%) | 2 (3.17%) | |

LHR, Laparoscopic Hartmann Reversal; OHR, Open Hartmann Reversal; ASA, American Society of Anesthesiologists; IQR, interquartile range

p-value: Independent t-test (T) or Wilcoxon rank sum test (W)

p-value: Chi-square test (C) or Fishers exact test (F)

## Discussion

Overall, the results of this study showed that LHR reduced POI frequency and decreased the time to solid diet and hospital stay compared to OHR. Similar results were observed in a group of patients who had previously undergone open Hartmann's operation in the subgroup analysis.

In colorectal surgery, the advantages of minimally invasive surgery, such as laparoscopic surgery, compared to open surgery include a more rapid recovery and fewer postoperative complications, such as wound infection. As such, minimally invasive surgery is strongly recommended by the 2018 ERAS Society [17]. However, LHR is more difficult than general laparoscopic surgery because of the presence of intra-abdominal adhesions caused by a past surgical history [6–10]. Nevertheless, the current literature on laparoscopic Hartmann's reversal highlights the advantages of the laparoscopic approach, such as reducing major complications after surgery and lowering the anastomotic leakage rate [18].

**Table 3. Operation related factors.**

| Variables | | Operation related factors | | |
|---|---|---|---|---|
| | | LHR (n = 63) | OHR (n = 63) | p-value |
| Stump length (cm) | | 15 (IQR 10–20) | 15 (IQR 10–20) | 0.9254(W) |
| Stump resection | Yes | 11 (17.46%) | 12 (19.35%) | 0.7846(C) |
| | No | 52 (82.54%) | 50 (80.65%) | |
| Adhesiolysis | Yes | 49 (77.78%) | 52 (82.54%) | 0.5028(C) |
| | No | 14 (22.22%) | 11 (17.46%) | |
| Combined resection | Yes | 9 (14.29%) | 10 (15.87%) | 0.8034(C) |
| | No | 54 (85.71%) | 53 (84.13%) | |
| Distance from anal verge to anastomosis (cm) | | 11 (IQR 8–15) | 15 (IQR 9–17) | 0.4869(W) |
| Anastomotic method | end-to-end | 54 (85.71%) | 53 (84.13%) | 0.8034(C) |
| | Other | 9 (14.29%) | 10 (15.87%) | |
| Circular stapler size (mm) | | 28 (IQR 25–28) | 28 (IQR 25–28) | 0.1771(W) |
| Diversion | Yes | 3 (4.76%) | 6 (9.52%) | 0.4915(F) |
| | No | 60 (95.24%) | 57 (90.48%) | |
| Surgery time (min) | | 210 (IQR 159–290) | 233 (IQR 160–280) | 0.7180(W) |

LHR, Laparoscopic Hartmann Reversal; OHR, Open Hartmann Reversal; IQR, interquartile range

p-value: Independent t-test (T) or Wilcoxon rank sum test (W)

p-value: Chi-square test (C) or Fishers exact test (F)

POI is a complication that not only causes patient dissatisfaction, but also prolongs hospital stay [19]. Furthermore, randomized controlled trials using scintigraphy and radiological transit studies to investigate the effect of laparoscopic colorectal surgery on gastrointestinal function confirmed that gastrointestinal function was improved following this surgery [20–22].

**Table 4. Postoperative outcomes.**

| Variables | | Postoperative outcomes | | |
|---|---|---|---|---|
| | | LHR (n = 63) | OHR (n = 63) | p-value |
| Length of stay (days) | | 9 (IQR 8–11) | 12 (IQR 10–16) | **<0.0001*(W)** |
| Time to solid diet (days) | | 4 (IQR 3–5) | 6 (IQR 5–8) | **<0.0001*(W)** |
| Clavien-Dindo classification | <IIIa | 60 (95.24%) | 58 (92.06%) | 0.7175(F) |
| | ≧IIIa | 3 (4.76%) | 5 (7.94%) | |
| Postoperative ileus | Yes | 3 (4.76%) | 14 (22.22%) | **0.0041*(C)** |
| | No | 60 (95.24%) | 49 (77.78%) | |
| Wound infection | Yes | 5 (7.94%) | 9 (14.29%) | 0.2568(C) |
| | No | 58 (92.06%) | 54 (85.71%) | |
| Anastomotic stricture | Yes | 0 (0.00%) | 2 (7.69%) | 0.2189(F) |
| | No | 29 (100.00%) | 24 (92.31%) | |
| Anastomotic leakage | Yes | 0 (0.00%) | 0 (0.00%) | NA |
| | No | 63 (100.00%) | 63 (100.00%) | |
| Intraabdominal abscess | Yes | 0 (0.00%) | 0 (0.00%) | NA |
| | No | 63 (100.00%) | 63 (100.00%) | |

LHR, Laparoscopic Hartmann Reversal; OHR, Open Hartmann Reversal

p-value: Independent t-test (T) or Wilcoxon rank sum test (W)

p-value: Chi-square test (C) or Fishers exact test (F)

LHR has also been reported to significantly reduce POI [13, 14]. Although these studies have the disadvantage of using single-center data, we once again confirmed that LHR plays an important role in reducing POI through propensity score matching using multi-center data.

In our study, LHR shortened both the time to soft diet initiation and the length of hospital stay. This finding agrees with that of a recent meta-analysis [18]. In addition, the recently revised guidelines from the American Society of Colon and Rectal Surgeons and the Society of American Gastrointestinal and Endoscopic Surgeons strongly recommend adopting a minimally invasive surgical approach because of its advantages in terms of bowel function and length of hospital stay during colorectal surgery [23]. Although LHR is more difficult to perform than general colorectal surgery, if surgeons with the requisite expertise are available, LHR should be performed as it is thought to be helpful in the early recovery of patients, such as speeding up time to solid diet, reducing postoperative pain, and shortening the length of hospital stay.

LHR is a challenging operation because of adhesions due to previous surgical history [6–10]. Accordingly, there are inevitable cases in which it is difficult to complete surgery with the laparoscopic approach. According to a meta-analysis, the average conversion rate is generally around 16.1%, but has been reported to reach up to 50% [24]. In our study, of a total of 104 patients who underwent the laparoscopic approach, 15 (14.4%) required conversion, showing results similar to those in the previous literature. Furthermore, when we analyzed the difference between LHR, OHR and conversion (S4 Table), the Conversion group had a longer hospitalization period (p = 0.016), but was not affected by POI (p = 0.1530) or time to solid diet (p>0.999). There was no difference in other complications. However, the length of stay in the conversion group did not differ from that of the OHR group (p>0.999), and there was no difference in operation time (p = 0.640) or other complications compared to the OHR group. This suggests that open conversion while using the laparoscopic approach does not affect the postoperative prognosis.

The primary strength of our study is that it was a multicenter propensity score-matched study. Although this was a retrospective study, post hoc power analysis using the observed differences in the primary outcome variable showed that the result was significant; moreover, the estimated power was confirmed to be approximately 82.6%. However, the present study had some limitations, including the inevitable inclusion of bias owing to the retrospective design. Second, there was a lack of description in the records regarding why the patients underwent Hartmann's operation and the degree of adhesion. It is thought that the specific reason such as severe comorbidity, elderly patients, proficiency of the surgeon for Hartmann's operation may be related to the degree of adhesion, which may affect the decision of the method of Hartmann reversal. Additionally, although the record was not shown, it is possible that selective bias occurred as laparoscopic Hartmann's operation was performed more often in the case of LHR. However, even in the subgroup analysis of the patient group who underwent Open Hartmann's operation, the advantages of LHR were similar to the study results of the entire patient group; therefore, our results support the current recommendations to proceed with the laparoscopic approach if possible.

## Conclusion

In conclusion, this study showed that LHR can be performed safely, and has the advantages of reducing POI and length of hospital stay compared with OHR. Because of the limitations of this retrospective study, a multicenter randomized study with long-term follow-up is needed to verify our results.

## Supporting information

**S1 Table. Baseline characteristics.**
(XLSX)

**S2 Table. Operation related factors.**
(XLSX)

**S3 Table. Postoperative outcomes.**
(XLSX)

**S4 Table. Baseline characteristics between LHR, conversion and OHR.**
(XLSX)

## Acknowledgments

It was supported by Uijeongbu St. Mary's Hospital Clinical Research Coordinating Center as part of the clinical trial activation project.

## Author Contributions

**Conceptualization:** Kil-yong Lee, Jaeim Lee.

**Data curation:** Kil-yong Lee, Jaeim Lee, Chul Seung Lee, Nam Suk Kim, Ju Myung Song, Ri-Na Yoo, Byung Jo Choi.

**Formal analysis:** Kil-yong Lee, Jaeim Lee.

**Investigation:** Kil-yong Lee, Jaeim Lee.

**Methodology:** Kil-yong Lee, Jaeim Lee.

**Project administration:** Kil-yong Lee, Jaeim Lee.

**Supervision:** Kil-yong Lee, Jaeim Lee, Seong Taek Oh.

**Validation:** Kil-yong Lee, Jaeim Lee.

**Visualization:** Kil-yong Lee, Jaeim Lee.

**Writing – original draft:** Kil-yong Lee.

**Writing – review & editing:** Kil-yong Lee, Jaeim Lee, Seong Taek Oh, Chul Seung Lee, Nam Suk Kim, Ju Myung Song, Ri-Na Yoo, Byung Jo Choi.

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
