## [Decision Letter · Decision Letter 0]

4 Apr 2023

PONE-D-23-05434

Postoperative effects of laparoscopic Hartmann reversal: A multicenter propensity score matched study.

PLOS ONE

Dear Dr. Lee,

Thank you for submitting your manuscript to PLOS ONE. After careful consideration, we feel that it has merit but does not fully meet PLOS ONE’s publication criteria as it currently stands. Therefore, we invite you to submit a revised version of the manuscript that addresses the points raised during the review process.

We look forward to receiving your revised manuscript.

Kind regards,

Yasunori Sato

Academic Editor

PLOS ONE

Journal Requirements:

Reviewers' comments:

Reviewer's Responses to Questions

**Comments to the Author**

1. Is the manuscript technically sound, and do the data support the conclusions?

Reviewer #1: Yes

Reviewer #2: Partly

2. Has the statistical analysis been performed appropriately and rigorously? 

Reviewer #1: Yes

Reviewer #2: No

3. Have the authors made all data underlying the findings in their manuscript fully available?

Reviewer #1: Yes

Reviewer #2: Yes

4. Is the manuscript presented in an intelligible fashion and written in standard English?

Reviewer #1: Yes

Reviewer #2: Yes

5. Review Comments to the Author

Reviewer #1: Title: Postoperative effects of laparoscopic Hartmann reversal: A multicenter propensity

score matched study.

The title is in line with study objectives and the results

Abstract: Structured and well written

Background: Detailed information on statement of problem as well as rational for the study clearly presented.

Methods: Fairly well described.

• Study design should be mentioned even though it is apparent from the methodology.

• Minimum sample size for the study was not calculated. This is desirable, as this will show whether the sample size in this study meets the minimum calculated sample size. If the sample size used is less than the calculated minimum sample size, then this can be a limitation in the study.

Result: Well written in details with relevant tables and figures

• The first column for most of the tables are without a label

• The state number of days for POI, hospital stay, etc are not really clear. Are these mean values or median? If it is mean value, please indicate it in the sentence and provide the standard deviation (SD) as well.

• Line 126 - Frequency cannot be in percentage. Please correct it. Is it the prevalence?

Discussion: The study findings are well discussed, with study limitations provided.

Conclusion: Clearly written with appropriate recommendation.

Reviewer #2: The author focused to the short-term outcomes of reversal Hartmann operation and compared the advantages of laparoscopic and opern surgery in this operation. They enrolled the patients from multi-institutions retrospectively and used the propensity score matched analysis for deep statistical analysis. From the analysis, the author have concluded that laparoscopic Hartmann reversal (LHR) is better method for fast recovery from the surgery and to prevent the postoprative bowel obstrution than open Hartmann reversal (OHR). Overall, the analysis seems to be fair and the results of this manuscript are similar to the previous studies and I agree with it. As the author pointed out, the study which reported the outcome of Hartmann reversal operation is not that many and these previous publications are mainly from single institution. This presented study is multi-instituional study with propensity score matced analysis and these background seems to strengthen the result compared to the previous studies. However, there are several concerns in this study which I want to point out.

1. As the author pointed out, the peritoneal adhesion will be the challenge to perform LHR. I feel that the conversion rate of LHR is another important outcome and I do not understand why the author excluded the 15 conversion cases from the analysis. It is more important to know how the LHR is difficult and conversion to open surgery may happen. Looking into the result, it seems that about 15% of patients were converted to open surgery and I recommend the author to analyze and discuss more deeply about it.

2. In Table 1, I see the data of the previous surrgery (Hartmann method) but this data is not presented in Table 2. This is very important imformation to consider the result of matching and I highly recommend to include it.

3. Hartmann operation will be considered when the anatomosis is considered to be difficult. However, reason of this decision will be various kinds of things; for example, perforation, severe comorbidity, elderly patients, profiency of the surgeon, etc. It would be better to include more specifc data why the patients underwent Hartmann operation at the beginning.

4. As the author mentioned in the limitation, there will be a selection bias since the patients with advanced cancer cannot undergo Hartmann reversal when they are under chemotherapy. It might be better to show how many patients underwent Hartmann surgery and how many had Hartmann reveersal.

5. The frequency of POI seems to be quite different. Is there any difference in the required treatment for these patient comparing LHR and OHR group? Were there any case which needed another surgery to treat POI? The primary outcome was set as POI and I recommend the author to analyze more deeply in it.

6. Although there was no significant difference, I am quite surprised that the incisional hernia was more frequent in LHR group. Why do you think this happened?

6. PLOS authors have the option to publish the peer review history of their article (what does this mean?). If published, this will include your full peer review and any attached files.

Reviewer #1: **Yes: **Prof. Tanimola Makanjuola Akande

Reviewer #2: No

---

## [Author Response · Author response to Decision Letter 0]

19 Apr 2023

Review Comments to the Author

Reviewer #1: Title: Postoperative effects of laparoscopic Hartmann reversal: A multicenter propensity

score matched study.

The title is in line with study objectives and the results

Abstract: Structured and well written

Background: Detailed information on statement of problem as well as rational for the study clearly presented.

Methods: Fairly well described.

• Comment: Study design should be mentioned even though it is apparent from the methodology.

We thank the reviewer for the considerate comments. Accordingly, we have indicated the study design at the beginning of the methods.

• Comment: Minimum sample size for the study was not calculated. This is desirable, as this will show whether the sample size in this study meets the minimum calculated sample size. If the sample size used is less than the calculated minimum sample size, then this can be a limitation in the study.

According to the reviewer’s comments, we have calculated sample size. As reported by Ng et al. [1] and Yang et al. [2], with 80% power and alpha 0.05, the sample size required for each group was 60 patients. Therefore, in this study, the number of samples included is appropriate, with a total of 126 patients and 63 patients in each group; the power was confirmed to be about 82% in the power analysis. This information has been added it in the “Patients” and “Results” section.

References

 1. Ng DC, Guarino S, Yau SL, Fok BK, Cheung HY, Li MK et al. Laparoscopic reversal of Hartmann's procedure: safety and feasibility. Gastroenterol Rep (Oxf). 2013;1(2):149-52. doi:10.1093/gastro/got018.

2. Yang PF, Morgan MJ. Laparoscopic versus open reversal of Hartmann's procedure: a retrospective review. ANZ J Surg. 2014;84(12):965-9. doi:10.1111/ans.12667.

Result: Well written in details with relevant tables and figures

• Comment: The first column for most of the tables are without a label

Accordingly, we have added the label to the first column.

• Comment: The state number of days for POI, hospital stay, etc are not really clear. Are these mean values or median? If it is mean value, please indicate it in the sentence and provide the standard deviation (SD) as well.

The number of days for hospital stay and the time to solid diet are described as the median. These have been added to the results section. 

• Comment: Line 126 - Frequency cannot be in percentage. Please correct it. Is it the prevalence?

This indicates the prevalence of POIs in each group. According to the reviewer’s comment, we have changed it to the number of occurrences as well as the percentage. 

Discussion: The study findings are well discussed, with study limitations provided.

Conclusion: Clearly written with appropriate recommendation.

 

Reviewer #2: The author focused to the short-term outcomes of reversal Hartmann operation and compared the advantages of laparoscopic and open surgery in this operation. They enrolled the patients from multi-institutions retrospectively and used the propensity score matched analysis for deep statistical analysis. From the analysis, the author have concluded that laparoscopic Hartmann reversal (LHR) is better method for fast recovery from the surgery and to prevent the postoprative bowel obstrution than open Hartmann reversal (OHR). Overall, the analysis seems to be fair and the results of this manuscript are similar to the previous studies and I agree with it. As the author pointed out, the study which reported the outcome of Hartmann reversal operation is not that many and these previous publications are mainly from single institution. This presented study is multi-instituional study with propensity score matced analysis and these background seems to strengthen the result compared to the previous studies. However, there are several concerns in this study which I want to point out.

1. Comment: As the author pointed out, the peritoneal adhesion will be the challenge to perform LHR. I feel that the conversion rate of LHR is another important outcome and I do not understand why the author excluded the 15 conversion cases from the analysis. It is more important to know how the LHR is difficult and conversion to open surgery may happen. Looking into the result, it seems that about 15% of patients were converted to open surgery and I recommend the author to analyze and discuss more deeply about it.

We appreciate the reviewer’s considerate comments. The initial intention of the group that underwent conversion was LHR; thus, if it were analyzed by intention-to-treat (ITT) method, this group should be analyzed by putting it in the LHR group. However, because the surgical method itself was open, it was excluded from our study. 

Nonetheless, according to the reviewer's opinion, we performed further analysis to investigate whether a conversion group differs from other groups. Although the surgical time was longer in this group compared to that in the LHR group, there was no difference in other postoperative complications and POI. We have added a supplement table summarizing the analysis of the three groups and have summarized the results in the discussion section.

2. Comment: In Table 1, I see the data of the previous surgery (Hartmann method) but this data is not presented in Table 2. This is very important information to consider the result of matching and I highly recommend to include it.

 � In the group that underwent open Hartmann's reversal, remarkably few cases were present where the previous operation was laparoscopic Hartmann's operation (3.9%); thus, we excluded Hartmann's method from matching in PSM and included this as a limitation in the discussion section. However, according to the reviewer's opinion, we have added the Hartmann method to Table 2 as well.

3. Comment: Hartmann operation will be considered when the anastomosis is considered to be difficult. However, reason of this decision will be various kinds of things; for example, perforation, severe comorbidity, elderly patients, proficiency of the surgeon, etc. It would be better to include more specifc data why the patients underwent Hartmann operation at the beginning.

 � Our study focused on Hartmann's reversal and investigated data from multi-institutional electric medical records. Furthermore, there were cases where Hartmann's operation was performed at institutions other than our six institutions, and there were no records from the other hospital; even though there was a surgical record for Hartmann's operation, there was no mention of the reason why Hartmann's operation was performed, and we had to exclude that item. Therefore, we have added this as a limitation of our study in the discussion section.

4. Comment: As the author mentioned in the limitation, there will be a selection bias since the patients with advanced cancer cannot undergo Hartmann reversal when they are under chemotherapy. It might be better to show how many patients underwent Hartmann surgery and how many had Hartmann reveersal.

According to the reviewer's opinion, the number of patients who underwent Hartmann's surgery in our institutes was investigated. In total, 1076 patients underwent Hartmann's operation in our hospital, and among them, 306 patients underwent Hartmann reversal. The remaining 31 patients who underwent Hartmann reversal in our institutes were patients who had undergone Hartmann's operation at other hospitals. This has been added to Figure 1.

5. Comment: The frequency of POI seems to be quite different. Is there any difference in the required treatment for these patient comparing LHR and OHR group? Were there any case which needed another surgery to treat POI? The primary outcome was set as POI and I recommend the author to analyze more deeply in it.

In the LHR group, two POI patients showed an improvement in symptoms at 2 and 3 days, respectively, after L-tube insertion with parenteral nutritional support. However, one patient underwent laparoscopic adhesiolysis with transverse colectomy to resolve adhesions.

Of 13 POI patients in the OHR group, three patients showed spontaneous improvement in their symptoms without L-tube insertion. However, one patient underwent open adhesiolysis with T-colostomy to resolve the adhesion. 

We have added these in the results section.

6. Comment: Although there was no significant difference, I am quite surprised that the incisional hernia was more frequent in LHR group. Why do you think this happened?

The exact cause is not known, but the occurrence of incisional hernia was confirmed by CT; since CT was performed in only about 38 out of 63 patients in each group, this may have resulted in a difference in incidence rates.

---

## [Decision Letter · Decision Letter 1]

9 May 2023

PONE-D-23-05434R1Postoperative effects of laparoscopic Hartmann reversal: A multicenter propensity score matched study.PLOS ONE

Dear Dr. Lee,

Thank you for submitting your manuscript to PLOS ONE. After careful consideration, we feel that it has merit but does not fully meet PLOS ONE’s publication criteria as it currently stands. Therefore, we invite you to submit a revised version of the manuscript that addresses the points raised during the review process.

We look forward to receiving your revised manuscript.

Kind regards,

Yasunori Sato

Academic Editor

PLOS ONE

Journal Requirements:

Reviewers' comments:

Reviewer's Responses to Questions

**Comments to the Author**

1. If the authors have adequately addressed your comments raised in a previous round of review and you feel that this manuscript is now acceptable for publication, you may indicate that here to bypass the “Comments to the Author” section, enter your conflict of interest statement in the “Confidential to Editor” section, and submit your "Accept" recommendation.

Reviewer #1: All comments have been addressed

Reviewer #2: All comments have been addressed

2. Is the manuscript technically sound, and do the data support the conclusions?

Reviewer #1: Yes

Reviewer #2: Yes

3. Has the statistical analysis been performed appropriately and rigorously? 

Reviewer #1: Yes

Reviewer #2: Yes

4. Have the authors made all data underlying the findings in their manuscript fully available?

Reviewer #1: Yes

Reviewer #2: Yes

5. Is the manuscript presented in an intelligible fashion and written in standard English?

Reviewer #1: Yes

Reviewer #2: Yes

6. Review Comments to the Author

Reviewer #1: Authors have addressed all the issues and suggestions raised in the earlier review quite satisfactorily.

Reviewer #2: Thank you for the correction. Most of my comments are fully addressed. One last comment from me is below.

1. I recommend the author to exclude the details for incisional hernia. Author have pointed out that almost half of patients did not have CT evaluation. Considering this fact, this result seems to be missleading.

7. PLOS authors have the option to publish the peer review history of their article (what does this mean?). If published, this will include your full peer review and any attached files.

Reviewer #1: **Yes: **Prof. Tanimola Akande

Reviewer #2: No

---

## [Author Response · Author response to Decision Letter 1]

10 May 2023

Reviewer #1: Authors have addressed all the issues and suggestions raised in the earlier review quite satisfactorily.

We thank the reviewer for reviewing the revised manuscript.

Reviewer #2: Thank you for the correction. Most of my comments are fully addressed. One last comment from me is below.

1. I recommend the author to exclude the details for incisional hernia. Author have pointed out that almost half of patients did not have CT evaluation. Considering this fact, this result seems to be missleading.

We thank the reviewer for the considerate comments. Accordingly, we have excluded the details of incisional hernia from the revised manuscript.

---

## [Editor Report · Decision Letter 2]

19 May 2023

Postoperative effects of laparoscopic Hartmann reversal: A multicenter propensity score matched study.

PONE-D-23-05434R2

Dear Dr. Lee,

We’re pleased to inform you that your manuscript has been judged scientifically suitable for publication and will be formally accepted for publication once it meets all outstanding technical requirements.

Kind regards,

Yasunori Sato

Academic Editor

PLOS ONE

---

## [Editor Report · Acceptance letter]

25 May 2023

PONE-D-23-05434R2 

Postoperative effects of laparoscopic Hartmann reversal: A multicenter propensity score matched study. 

Dear Dr. Lee:

I'm pleased to inform you that your manuscript has been deemed suitable for publication in PLOS ONE. Congratulations! Your manuscript is now with our production department. 

Kind regards, 

on behalf of

Dr. Yasunori Sato 

Academic Editor

PLOS ONE